# The Increasing Impact of Translational Research in the Molecular Diagnostics of Neuromuscular Diseases

**DOI:** 10.3390/ijms22084274

**Published:** 2021-04-20

**Authors:** Dèlia Yubero, Daniel Natera-de Benito, Jordi Pijuan, Judith Armstrong, Loreto Martorell, Guerau Fernàndez, Joan Maynou, Cristina Jou, Mònica Roldan, Carlos Ortez, Andrés Nascimento, Janet Hoenicka, Francesc Palau

**Affiliations:** 1Department of Genetic and Molecular Medicine—IPER, Hospital Sant Joan de Déu and Institut de Recerca Sant Joan de Déu, 08950 Barcelona, Spain; dyubero@sjdhospitalbarcelona.org (D.Y.); jarmstrong@sjdhospitalbarcelona.org (J.A.); LMartorell@sjdhospitalbarcelona.org (L.M.); gfernandezi@sjdhospitalbarcelona.org (G.F.); jmaynou@sjdhospitalbarcelona.org (J.M.); mroldanm@sjdhospitalbarcelona.org (M.R.); 2Center for Biomedical Research Network on Rare Diseases (CIBERER), ISCIII, 08950 Barcelona, Spain; anascimento@sjdhospitalbarcelona.org; 3Neuromuscular Unit, Department of Pediatric Neurology, Hospital Sant Joan de Déu and Institut de Recerca Sant Joan de Déu, 08950 Barcelona, Spain; dnatera@sjdhospitalbarcelona.org (D.N.-d.B.); ciortez@sjdhospitalbarcelona.org (C.O.); 4Laboratory of Neurogenetics and Molecular Medicine—IPER, Institut de Recerca Sant Joan de Déu, 08950 Barcelona, Spain; jpijuan@fsjd.org; 5Department of Pathology, Hospital Sant Joan de Déu, Pediatric Biobank for Research, Institut de Recerca Sant Joan de Déu, 08950 Barcelona, Spain; cjou@sjdhospitalbarcelona.org; 6Confocal Microscopy and Cellular Imaging Unit, Institut de Recerca Sant Joan de Déu, 08950 Barcelona, Spain; 7Division of Pediatrics, Clinic Institute of Medicine & Dermatology, Hospital Clínic, University of Barcelona School of Medicine and Health Sciences, 08950 Barcelona, Spain

**Keywords:** genetic diagnostics, molecular diagnostics, multi-omics, neuromuscular diseases, translational diagnostics, translational research

## Abstract

The diagnosis of neuromuscular diseases (NMDs) has been progressively evolving from the grouping of clinical symptoms and signs towards the molecular definition. Optimal clinical, biochemical, electrophysiological, electrophysiological, and histopathological characterization is very helpful to achieve molecular diagnosis, which is essential for establishing prognosis, treatment and genetic counselling. Currently, the genetic approach includes both the gene-targeted analysis in specific clinically recognizable diseases, as well as genomic analysis based on next-generation sequencing, analyzing either the clinical exome/genome or the whole exome or genome. However, as of today, there are still many patients in whom the causative genetic variant cannot be definitely established and variants of uncertain significance are often found. In this review, we address these drawbacks by incorporating two additional biological omics approaches into the molecular diagnostic process of NMDs. First, functional genomics by introducing experimental cell and molecular biology to analyze and validate the variant for its biological effect in an in-house translational diagnostic program, and second, incorporating a multi-omics approach including RNA-seq, metabolomics, and proteomics in the molecular diagnosis of neuromuscular disease. Both translational diagnostics programs and omics are being implemented as part of the diagnostic process in academic centers and referral hospitals and, therefore, an increase in the proportion of neuromuscular patients with a molecular diagnosis is expected. This improvement in the process and diagnostic performance of patients will allow solving aspects of their health problems in a precise way and will allow them and their families to take a step forward in their lives.

## 1. Introduction

Inherited neuromuscular disorders (NMDs) are a group of diseases that affect the muscle and the peripheral nervous system (PNS). The severity varies widely, from mild impairment of voluntary movements to complete paralysis [1]. The diagnosis of NMDs is based on the appropriate combination of clinical and neurological examinations, electrophysiological studies of the peripheral nervous system, histopathology of the muscle biopsy (and sometimes of the nerve biopsy), and gene/genome studies. This clinical approach is especially relevant to address difficult differential diagnoses and complex phenotypes or diseases [2]. Inherited NMDs vary in their onset from the prenatal period to adulthood but mainly affect infants and children. According to the GeneTable of Neuromuscular Disorders (http://www.musclegenetable.fr/ (accessed on 20 January 2021)) [3], by December 2020, almost 600 genes had been associated with NMD. This picture includes disorders of the PNS in the strict sense, but also hereditary ataxias, spastic paraplegia and cardiomyopathies.

Next-generation sequencing (NGS), which enables the simultaneous analysis of many genes, is particularly useful in NMDs, given their high clinical and genetic heterogeneity and the extraordinary length of many of the involved genes [4,5]. Implementation of NGS into routine clinical practice has dramatically changed the diagnostic algorithms applied in patients with NMDs. Nonetheless, clinical evaluation, neurophysiological and histopathological information is still essential for precision phenotyping to select the most appropriate genetic test for individuals with NMDs. The most obvious example of the decisive role of clinical evaluation is simple clinical observations that act as reliable clues to using single-gene testing as a first-line approach in some of the most frequent disorders. The primary role of biomarkers in guiding the genomic strategy of NGS is exemplified in most inherited metabolic disorders, which have biochemical biomarkers that reduce the number of candidate genes.

Here we illustrate the current scenario of genetic diagnosis in NMDs by reviewing molecular and genomic techniques and approaches, their indications, benefits and limitations. In addition, we describe novel approaches that are solving problems arising from NGS analysis in a considerable number of NMDs patients.

## 2. From the Classical Molecular Approach to the Genome Scenario

Until a few years ago, the path to diagnosis began with clinical evaluation, continued with complementary examinations (laboratory tests, electromyogram, muscle biopsy, muscle MRI, etc.) and, finally, an individual candidate gene was chosen to be studied by molecular tests or Sanger sequencing. Nowadays this process has changed significantly. The clinical evaluation continues to be the initial step and the basis for determining the genetic studies to be performed. If there are clinical clues that point to a specific disease with a single causal gene, single-gene testing is recommended (Table 1) [6,7,8,9,10,11,12]. On the other hand, when the suspected disease belongs to a group in which many genes are known to be involved (e.g., congenital myopathies, congenital myasthenic syndromes, limb-girdle muscular dystrophies, peripheral neuropathies, etc.), NGS-based testing is the most convenient option [13,14,15,16]. Again, in many cases neurophysiological, histological and laboratory studies are still very important approaches to confirm or rule out genetic findings or investigate complex phenotypes.

Over the past 10 years, NMDs genetic analysis has moved from gene sequencing to multigene screening using NGS, which includes clinical exome sequencing (CES), whole exome sequencing (WES) and whole genome sequencing (WGS). A recent survey by the EURO-NMD European Reference Network showed that NGS is the most common technical diagnostic approach in the European referral centers (more than 50% of all genetic tests performed), with the gene panel testing being the strategy used by two-thirds of centers, followed by WES [17]. WGS also has enormous potential for diagnostics [18], but is still moving from research to standardized use in clinical practice.

Many groups have reported their genetic diagnostic strategies for NMDs, with diagnostic yields ranging from 20 to 70% [19,20,21,22,23,24,25,26]. Diagnostic rates using NGS are often difficult to compare due to differences not only in genetic strategy but also in cohort selection criteria. The highest diagnostic rate reported (73%) was achieved in a cohort of 45 consanguineous NMD patients [27,28]. Diagnostic rates of NGS also vary substantially depending on the disease subgroup. In general, the diagnostic rates obtained in cohorts of patients with muscle and myasthenic conditions are higher (≈60%) [2,29,30] than the rates in pre-junctional conditions. The diagnostic yield of NGS in a series of 133 patients followed in our unit with paraparesis and neuropathy was 25% (after exclusion of PMP22 duplication/deletion in those patients with typical demyelinating CMT). In Charcot-Marie-Tooth disease and other genetic neuropathies, rates vary from 9 to 47% [20,31,32,33,34], subject to the cohort collection and variant classification. Achieving a genetic diagnosis is more likely in patients with an early onset and a demyelinating neuropathy [20].

Deep-phenotyping using the standardized Human Phenotype Ontology (HPO; https://hpo.jax.org/app/) terminology [35] has proven to be useful for filtering, prioritizing and interpreting genetic variants detected by NGS [36]. To obtain a detailed phenotype is required an optimal clinical assessment and, depending on the case, histopathological, neurophysiological and muscle MRI studies may also be necessary. This, in addition to the fact that most NMD lack clinical signs that point to a specific gene but rather to a group of genes, confirms the usefulness of placing genetic and genome studies at the first line of the diagnostic process, relegating other tests to evaluate the possible uncertainty of genetic findings, or even to reaffirm the validity of pathogenic variants.

Although advances in NGS have greatly facilitated NMD diagnosis, overall genetic diagnostic rates remain below 60%. A part of the unsolved NMD cases can probably be explained by complex genetic mechanisms, such as a digenic or oligogenic inheritance, causative mosaic variants, but also by cryptic splice mutations that are not identified. In addition, there are technical restrictions of NGS-based genome analyses. Regarding WES limitations, this approach has (i) the inability to reliably detect structural variations in DNA, including copy number variants such as intragenic deletions or duplications, structural rearrangements, and tandem-repeat expansions and (ii) the inappropriateness to adequately capture intronic and regulatory regions [4,37]. On the other hand, WGS can detect variants in both coding and non-coding regions but the interpretation of the results remains challenging. It is generally performed on a research basis, and few cases are found in the literature. A case illustrating the perfect utility of WGS is the finding of a five megabase inversion involving the *DMD* gene [38], which was validated with RNA sequencing from the patient’s muscle. Although WGS will become gradually adopted in the clinical setting for rare disease diagnosis, targeted sequencing of Mendelian genes in Online Mendelian Inheritance in Mam (OMIM, https://www.omim.org) by CES and/or WES is still valuable for deeper unraveling the genetic complexity in specific regions. Therefore, a combination of CES, WES, and WGS is recommended, which may augment the clinical utility of NGS-based genetic testing [39]. Particular knowledge is required upon certain genes, for example, regarding giant genes, particularly frequent in NMDs, there are technical considerations that modify the prediction of the variant impact on the gene or protein level [5]. Large multiexonic genes like *TTN* [40] and *NEB* undergo extensive alternative splicing events in different developmental and physiological states, with different factors and proteins regulating these alternative splicings, using this knowledge for a more accurate evaluation of mutations [5]. Another interesting fact is that, as in the case of titinopathies, the different isoforms show specific progression-related patterns of muscular involvement [41], requiring a deep phenotyping description to be matched with the genotype. Yet, another complication is the identification of copy number variants from high throughput data.

## 3. Gene Gathering—The Path to Phenotype

The identification of NMD genes has exponentially increased. An average of 27 genes and 21 new phenotypes have been identified annually in the last 4 years, according to the Gene Table of Neuromuscular Disorders [3]. In pediatric medicine is important to consider the various stages of the child’s development and how these may affect the natural history of the disease. At the time of the first medical visit or the diagnostic suspicion, the phenotype in the child may be incomplete, since NMDs have a variable age of onset. Most of the genes included in the neuromuscular Gene Table are listed in more than one of the 16 disease groups, illustrating the genetic heterogeneity and wide phenotypic spectrum inherent to NMDs. For example, very early stages of mild/late-onset motor neuropathies may lead to misclassification with other types of neuromuscular disorders [42]. Some patients with nonsense mutations in NEFL who expressed both patterns [43] illustrate the overlap between neuropathic and myopathic expression. This panorama exposes a scenario in which it should be suspected that there are genes that can cause different diseases that affect different structures of the peripheral nervous system. This supposes an extension of the lists of genes reported for the different groups of NMDs.

Proper selection of genes to be analyzed is critical in targeted NGS. Some efforts have been made to establish the lists of genes to be analyzed in the diverse NMDs. The aim is to apply a cost-effective strategy that enables to find of the causative variants and limits the number of incidental findings. Recently, a national French consensus has been published, setting up a strategy for NGS analyses in myopathies. It is based on a successive NGS analysis of a “core gene list” defined for each one of the 13 clinical and histological diagnosis groups of myopathies, and followed in case of negative results by the analysis of an “exhaustive gene list” [44]. The development of freely available databases (NeuroMuscleDB; http://yu-mbl-muscledb.com/NeuroMuscleDB/) and tools to facilitate the design of gene panels by experts (e.g., PanelApp; https://panelapp.genomicsengland.co.uk) have proven to be helpful [45,46]. One of the most useful strategies to prioritize genes is the implementation of semantic ontologies, controlled vocabularies with a systematic structure that allows easy data retrieval and analysis. The Human Phenotype Ontology (HPO) is a worldwide standard to describe and computationally analyze phenotypic abnormalities found in human disease, assisting to link genes and clinical signs [47]. Phenotype-based filtering and prioritization contribute to the interpretation of genetic variants detected in exome sequencing and may be even more efficient than curated disease-gene virtual panels in case of misinterpretation of the disease group by the clinicians [36]. In our experience, as an academic pediatric center with specific programs on rare diseases, the application of HPO retrieved highly beneficial results for diagnosis, especially on patients with complex neurodevelopmental phenotypes. Since the introduction of NGS testing for genetic rare diseases in 2017, the global diagnosis rate of suspected single-gene diseases has increased and currently is 33%. Regarding NMDs, the molecular diagnosis is based either on the directed genetic test suggested by the neurologist based on the clinical diagnosis (Table 1), or NGS (CES or WES) is performed together with the HPO profile. Of the 86 neuromuscular patients for whom we combined exome studies and HPO codes, we found causal pathogenic variants in 17% of patients, and variants of unknown significance were retrieved in 33% requiring a further evaluation to determine the disease causality. Even though most of the genes could be found also using gene lists, some of the findings arise only due to the HPO filtering pipeline. A 2018 study adopted a strategy for identifying the phenotype-influencing variants coming up from any of the terms from the HPO database pointing to muscle physiology or structures [48]. However, missing variants located in genes not yet associated with muscle disease, or missing variants coding for an interactome of the known causative proteins was an issue [48]. 

The goal is to find the causative variants and to verify all the variations for which an altered function is presumed that may produce a phenotype similar to that of our patient, or reflect a congruence between phenotype and genotype. The powerful impact NGS technology has provided in NMDs was unimaginable one decade ago. The analysis has become a multidisciplinary task in which laboratory geneticists, clinical scientists and bioinformaticians, neurologists, clinical geneticists, and genetic counselors require a strong interaction. The knowledge requisites for the interpretation have increased and other factors like data deciphering and management appear as daily issues. Massive data cannot be solely analyzed, it requires an upgrade of the dimensional view, aside from incorporating population databases, pathogenicity scores and prediction tools, protein domains and interactions analysis, and it demands a broad vision of the genome, integrating regulatory epigenetic data, RNA and DNA interactions, to unravel other uncertain mechanisms behind pathology. Combining and integrating massive genomic data is essential to match the genotype with a phenotype with an expanded view, to bring them closer together.

## 4. The Remaining Unsolved

Despite all the efforts, a large proportion of patients with NMDs remain undiagnosed. Some causative variants might go unnoticed due to non-uniform coverage of gene sequence in the DNA, but the main problem is most likely methodological. Regarding sequencing, targeted NGS does not examine genes or genomic regions not incorporated in the panel, while WES does not interrogate depth intronic regions, intragenic regions, regulatory regions, pseudogenes or structural variants [49]. In contrast, WGS is a good choice for detecting small copy number variants and other structural variations, but as with the other NGS approaches, it does not accurately resolve DNA expanded dynamic mutations [50], which would require longer read NGS technology. On the other hand, some of the structural muscle genes are giant and contain repeat regions such as NEB or TTN, resulting in poorly called regions [51]. These genes’ characteristics would require a specialized classification and analysis to improve NGS interpretations and candidate selection [5]. In these particular cases, long reads would work better in the quality of examining these genes, not only to avoid missing variants but also to detect intragenic CNVs. 

Another factor to consider when analyzing genes in undiagnosed NMD patients is the mode of inheritance since non-Mendelian [52] or digenic inheritance [53], which demand different methodological strategies, are also involved. Additionally, cases with somatic mosaicisms require deeper coverage sequencing to be detected [51].

## 5. Conflicts with Uncertainty 

One of the main challenges unrelated to the NGS methodology is the interpretation of the pathogenicity of variants. During the analysis and interpretation of NGS data in a particular patient, a large amount of genomic data is generated and it is frequent to find variants of uncertain significance. Based on genetic, population frequency and predicted functional criteria, the American College of Medical Genetics and Genomics (ACMG) and the Association for Molecular Pathology (AMP) have elaborated standards and guidelines to classify genetic variants into five categories: benign, likely benign, variant of uncertain significance (VUS), likely pathogenic, or pathogenic [54]. The Clinical Genome Resource (ClinGen) Sequence Variant Interpretation (SVI) Working Group that aims to standardize the application of the ACMG/AMP guidelines is providing recommendations for an improvement in the prediction of pathogenicity of variants (https://clinicalgenome.org/working-groups/sequence-variant-interpretation) Following the ACMG guidelines, there are different bioinformatic tools to facilitate the classification process (e.g., Varsome, https://varsome.com/; Franklin, (https://franklin.genoox.com/clinical-db) Intervar, (http://wintervar.wglab.org) though conflicting interpretations might arise due to the use of different base criteria. Thus, aside from the clinical, familiar history and segregation data of the patient, each geneticist must provide specific knowledge and internal consensus criteria to be able to classify a certain variant. This is exemplified with disease databases, for instance: a comparison of the classifications submitted on ClinVar performed to understand the enormous differences that we might find when assessing a certain variant (conflicting interpretations of pathogenicity) [55]. 

Facing uncertainty has become the daily battle for geneticists and physicians, increasing the challenges for genetic counseling and sometimes the proper treatment. Despite the differences of interpretation in different studies, the ratio of patients with VUS that could explain the phenotype is similar or higher than the ratio of patients with pathogenic variants that allow molecular diagnosis [56]. In some occasions, familiar studies can clarify doubts (e.g., by demonstrating that the uncertain variant generates a de novo mutation, which modifies the significance to likely pathogenic or pathogenic), but those variants that are not reclassified remain uncertain until further studies permit to prove its pathogenicity or benignity. Some genetics departments and laboratories close the cases by reporting the VUS, but the most common strategy is to re-evaluate the finding at defined time intervals [17]. Matchmaking has emerged as a helpful tool to improve diagnostic efficiency [57] through the exchange of data. Finding another case using this method can be very useful and is becoming a tool for clinical and scientific use. Importantly, the discovery of biomarkers, or the setting up of functional validation of variants in the clinical setting are some of the needs to improve what has already been called the diagnostic odyssey.

The phenotype can also be a source of uncertainty, since a significant number of genes can have pleiotropic effects and different phenotypic consequences, generating unexpected inconsistencies. Two illustrative examples are the novel association of hnRNPA2B1 (previously associated with frontotemporal dementia, amyotrophic lateral sclerosis, inclusion body myopathy, and/or distal myopathy) with early-onset oculopharyngodistal muscular dystrophy [58] and SCN4A (a well-known cause of myotonia and periodic paralysis) with congenital myopathy [59].

## 6. What to Do—The Functional Genomics Scenario

Genetic/genomic testing aims to generate an etiological diagnosis based on objective data to support medical decision-making. In the clinical setting, genetic testing can identify or confirm the cause of the disease and thus assist in making therapeutic decisions. However, despite the efforts, there are undiagnosed patients [60,61] and the diagnostic deficit [62,63] is still a major problem in managing individuals with a suspected rare disease. To address this deficit and the diagnostic odyssey [64], the implementation of strategies focused on undiagnosed rare diseases (URD) is a growing approach that is being incorporated in referral academic hospitals and centers [60,61,65]. These strategies should combine the clinical precision phenotyping and experimental approaches. In our hospital, we have developed the in-house pipeline Translational Diagnostics Program (TDP) to functional validate genetic variants whose clinical significance is unknown [66,67]. In the neuromuscular setting, such a program requires multidisciplinary teamwork. 

The TDP requires close contact between clinicians with expertise in neuromuscular disorders, geneticists, clinical scientists in diagnostic laboratories, pathologists and research-based investigators to determine the pathogenicity of genetic variants. The variants classified as VUS makes clinical decision-making difficult and experimental methods are helpful to reclassify these variants from VUS to pathogenic or benign. To assist in the process of reclassifying variants concerning a patient’s phenotype and to improve the diagnostic deficit, a focus on the frontiers of medicine and scientific research [66] is required in a patient-centered model of medicine and healthcare to solve the diagnosis [62,68]. With this proposal in mind, we and others are using different tools of experimental and computational biology to analyze VUS and determine the function and possible pathophysiological alteration of the encoded protein [66,69]. The objective is to delineate the impact of VUS using a holistic approach based on the triangle “precision phenotyping—clinical genomics—functional genomics”. This program combines four stages [66]: (i) in-depth and precision phenotype evaluation to define HPOs; (ii) clinical genomics with information of the variant identified in a candidate gene; (iii) functional genomics, for the validation of the genetic variant through molecular and cellular experiments; and (iv) making diagnostic decisions by the referring physician or the clinical team based on the recommendations of the potential pathogenic variant (Figure 1). Using this pipeline in NMDs patients, we have demonstrated the effectiveness of the TDP pipeline to resolve clinical doubts when the genetic variant detected by NGS is a VUS or when the relationship between phenotype and genotype is not congruent or partial [66].

For example, we have validated the p.R1623W variant of the *DYNC1H1* gene (MIM *600112, Dynein 1) as the cause of the complex clinical phenotype observed in a patient showing a partial phenotype correlation with all *DYNC1H1*-related phenotypes reported in OMIM. We used recombinant DNA techniques to generate GFP-tagged *DYNC1H1* wild-type and variant constructs to compare the subcellular localization of the recombinant proteins. In transfected SH-SY5Y neuroblastoma cells, we demonstrated that p.R1623W dramatically affects the localization of cytoplasmic dynein 1 (Figure 2A), the protein encoded by *DYNC1H1*. These findings suggest that p.R1623W is a pathogenic variant and therefore it is the cause of the complex clinical phenotype observed in the patient [66]. 

We are also using the patient’s fibroblasts as a cell model to investigate the impact of a given candidate variant after NGS sequencing upon the encoded protein. For instance, we studied a patient carrying the p.K75E variant in the *DNM1L* gene (MIM *603850, DRP1). *DNM1L* gene mutations have been associated with lethal encephalopathy and optic atrophy 5, however, the patient showed spastic paraparesis, not previously associated with *DNM1L*. The experimental studies in the patient’s fibroblasts revealed an aberrant mitochondrial network like other *DNM1L*-associated cases (Figure 2B). These findings supported the pathogenicity of p.K75E in *DNM1L* and revealed a novel relationship between this gene and hereditary spastic paraparesis that was part of the particular phenotype in this patient [66]. 

Our current results show that the TDP is feasible and it can improve diagnostic performance when the variant found after NGS sequencing is a VUS. We have presented examples where the VUS is a missense variant. It should be noted that TDP could also be applied in the case of genetic variants that affect consensus splice regions and even regulatory regions of gene expression. For VUS affecting canonical splice sites, a splicing study can be performed using minigenes by cloning the genomic region [55] or by analyzing RNA from a patient’s fibroblasts by RT-PCR. These two strategies examine the impact of a given VUS on the splicing and provide information on the transcripts affected by the variant. For VUS of regulatory regions, cloning of the genomic region for studies of luciferase activity could be an option for the evaluation of the functional impact on the transcriptional control of gene expression.

However, much remains to be done to be able to reach the diagnosis of all NMDs patients. The incorporation within referral hospitals or hospital-associated research institutes of other experimental approaches, including omics and animal models either invertebrates (*Caenorhabditis elegans*, *Drosophila melanogaster*) [70] or vertebrates (*Danio rerio*), as part of the diagnostic activity, is a necessary step that academic healthcare centers can take to provide a diagnosis when the result of an exome/genome did not yield a diagnosis. Network-based approaches [61] are a realistic way to address the diagnostic deficit in NMDs, but despite being a challenging task, the development of hospital-based TDPs is also a step forward in developing the patient-centered healthcare paradigm in precision medicine. 

## 7. Multi-Omics Approaches

While healthcare laboratories focus on NGS benefits, solving unsolved issues remains the essential need. Massively parallel technologies appear as powerful approaches that, integrated with sequencing and phenotypic data, are compelling for diagnosis. Different biological molecules can be addressed with omics techniques beyond the DNA, such as those related to the study of RNA sequence, metabolites and proteins. They offer complementary data that could help to understand the pathological context of a specific patient. The incorporation of these powerful omics approaches into the diagnostic process still needs additional efforts.

Recently, some groups have proven the utility of analyzing the patient’s transcriptome for rare Mendelian disease. Performing RNA sequencing (RNA-seq) on patients with negative results in WES or targeted NGS yielded a diagnosis in 35–36% of the cases in NMD cohorts [38,71]. RNA sequencing allows detecting aberrant expression, aberrant splicing and mono-allelic expression. Some of the challenges of using this technology reside in sample variability due to different factors. For instance, tissue selection for RNA sequencing is particularly relevant because gene expression is expected to be different according to the disease target tissue. For NMDs, it has been determined that myotubes generated by transdifferentiation from an individual’s fibroblasts accurately reflect the muscle transcriptome and faithfully reveal disease-causing genes [38]. This work provides support for the utility of the transcriptome study for the detection of pathogenic variants associated with neuromuscular disorders. Additionally, artificial intelligence approaches can enhance our ability to find genes, improve functionality predictions and may improve the capacity of molecular diagnostics [72].

The analysis of proteins and the proteome, their activity, interaction, localization and composition, structure and function are some of the aspects proteomics may resolve. Proteomic experiments are generally used to detect specific signatures of disease progression [73], to unravel pathogenicity mechanisms and detect biomarkers, or to discover targets or processes for therapeutic intervention [74,75]. In 2004, Jaffe et al. coined the concept of proteogenomics, a viewpoint focused on the integration of genomic and proteomic data [76]. The idea is to obtain a robust validation of the consequences of genomic findings, which provides evidence of the effects of the genetic defect. Applied proteogenomics facilitates the assessment of the consequences of genetic variation on protein abundances (copy-number variations, single nucleotide variants (missense or nonsense), 5′ and 3′ untranslated regions, splice-site mutations and programmed frameshifts) [77]. However, a limitation of the approach is that many low-abundant proteins might fail to be identified [77].

Although metabolomic profiles have been used as biomarkers for disease progression and response to treatment, they are now increasingly applied in diagnostics to determine the functional consequences of variants of unknown significance in NGS. Inherited metabolic disorders are caused by defects in metabolic pathways resulting in a deficiency of intermediates and a build-up of toxic metabolites. In mitochondrial disease, analysis of blood metabolites can distinguish between many conditions, including progressive external ophthalmoplegia, mitochondrial myopathy, or mitochondrial encephalomyopathy, and lactic acidosis or stroke-like episode. The identification of biomarkers like GDF-15 in the transcriptome of skeletal muscle from TK2-associated patients raises the possibility of metabolic fingerprinting for mitochondrial diseases [78]. Metabolic signatures identified for muscular dystrophies and Charcot-Marie-Tooth disease type 1A could be used to indicate disease severity and aid diagnosis [79,80,81]. 

Stenton et al. suggested a multi-omics approach to the diagnosis of Mendelian disease [82] in which genomic data obtained from WES and WGS negative studies are analyzed in parallel with transcriptomic and proteomic data. Therefore, genomics together with transcriptomics (detection of aberrant transcript expression, aberrant splicing and mono-allelic expression events) plus proteomics (detection of aberrant protein expression), provides evidence to find causative variants at the genomic level. In addition to omics studies, it is interesting from the phenotype perspective to find new approaches that use systems biology to produce functionally coherent groups of phenotypes that provide a new perspective on cellular functions and phenotypic patterns that underlie NMDs [83]. In summary, the integration of omics into the diagnosis pipeline is expected to increase the diagnostic yield of unsolved cases by WES or WGS.

## 8. Concluding Remarks

The diagnostic process of NMDs requires a clinical and biological evaluation of the patient. This process includes the definition of the phenotype and clinical expression, the natural history according to the age of the patient, the family history, and the complementary tests, where genetic testing stands out along with the use, on the indicated occasions, of the electrophysiological exams, histopathological tests and MRI imaging. However, two aspects still make etiological diagnosis difficult in a significant number of patients in whom a genetic NMD is suspected: (i) first, the increase in genes associated with recognized phenotypes and nosological entities, and the overlapping of phenotypes; and (ii) second, the difficulty that sometimes arises as a result of the uncertainty in the pathogenic significance of the genetic variant(s) in a candidate gene, or the incongruity between genotype and phenotype despite finding a pathogenic variant. The first question has been answered through the systematic application of NGS-based genetic testing, either CES, WES, WGS, or RNA-seq. The second issue requires the implementation of translational diagnostic programs that work with patient cells (e.g., fibroblasts) and/or invertebrate and vertebrate animal models, which incorporate experimental biology in the diagnostic process in centers of expertise and tertiary referral hospitals. Multi-omics approach is allowing the incorporation of transcriptomic and proteomic biomarkers in the molecular diagnosis of NMDs and monitoring of the therapeutic response to new treatments. The set of both proposals, translational research and omics, will increase the diagnosis of patients affected by a neuromuscular disorder, influencing the model of the diagnostic process in the referral and academic centers.

## Figures and Tables

**Figure 1 ijms-22-04274-f001:**
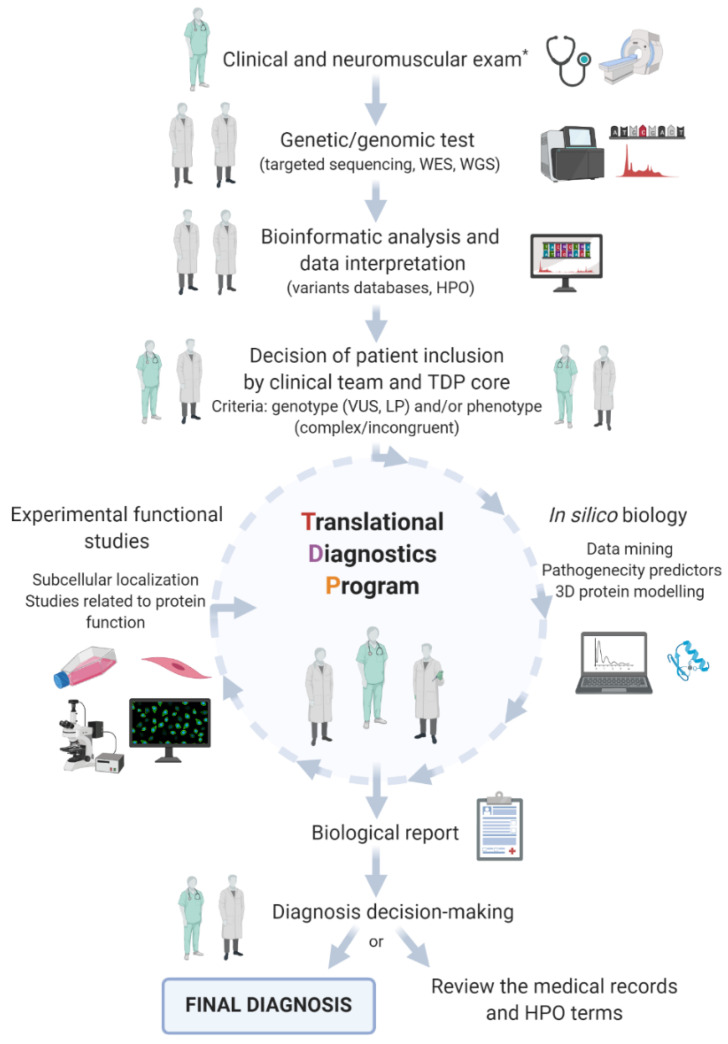
*In-house Translational Diagnostics Program (TDP) flow diagram in neuromuscular diseases*. First of all, a precision phenotyping by a clinical and neuromuscular exam is performed in the patients. After, the most appropriate genetic/genomic test is selected. The bioinformatics analysis and data interpretation are carried out using a pipeline developed in our department. In this step, if diagnosis is not achieved, the clinical team and the TDP core team decide the patient’s inclusion in the study into the TDP. When the case is accepted, firstly, in silico biology of the variant are performed (literature review and data mining, pathogenicity predictors, and 3D-protein modelling). Secondly, functional validation studies are performed by reliable molecular, cellular, and imaging assays related to specific protein function. After that, a final biological report is generated integrating both in silico and experimental results. In the final step, the clinical team and TDP core team evaluate all procedure steps and elaborate the final diagnostic decision. Abbreviations: HPO, Human Phenotype Ontology; LP, likely pathogenic; VUS, variant of uncertain significance; WES, whole exome sequencing; WGS, whole genome sequencing. * Anamnesis, physical examination, biomarkers, neuroimaging and pathology. Illustration created with BioRender (https://biorender.com).

**Figure 2 ijms-22-04274-f002:**
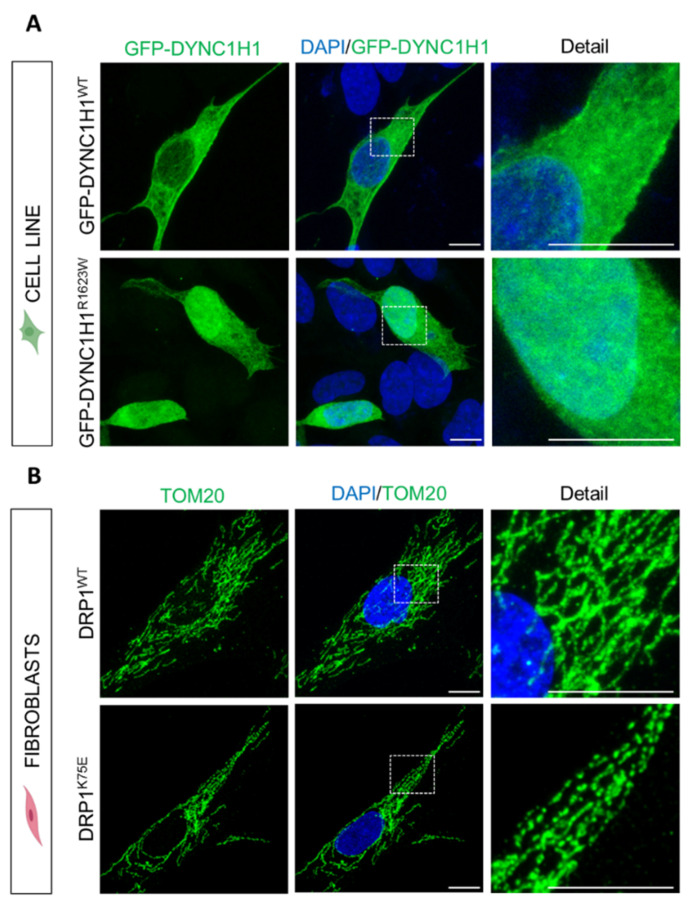
Experimental functional studies in patients with genetic variants in *DYNC1H1* and *DNM1L* (encodes DRP1) using a cell line model and patient’s fibroblasts. (**A**) Cellular expression pattern of GFP-*DYNC1H1*^WT^ (top panels, green, wild-type) and GFP-*DYNC1H*1^R1623W^ (bottom panels, green, patient’s variant) in SH-SY5Y-transfected cells. GFP-*DYNC1H1*^WT^ was observed in the cytoplasm, whereas GFP-*DYNC1H1*^R1623W^ was abnormally delocalized at the nuclei. (**B**) Mitochondrial network structure stained with TOM20 (green) and nucleus (DAPI; blue) in fibroblasts of control (top panels, DRP1^WT^) and patient (bottom panels, DRP1^K75E^). Patient’s fibroblasts showed an abnormal mitochondrial network chain-like structure. The magnification of the withe dashed boxes regions are shown in the right panels (detail). Scale bar: 10 µm.

**Table 1 ijms-22-04274-t001:** Neuromuscular diseases in which genetic testing of a single-gene approach is recommended.

	MLPA	CMA	NGS	TP-PCR	Sanger	Southern Blotting	Long-Range PCR
Dystrophinopathies ^a^ [6,7]	1	-	2	-	3	-	-
5q-linked SMA ^b^ [8]	1	-	-	-	2	-	-
DM1 and DM2 ^c^ [9]	-	-	-	1	-	2	-
FSHD1 and FSHD2 ^d^ [10]	-	-	-	-	2	1	-
CMT1A dup and HNPP del ^e^ [11]	1	1	-	-	-	-	-
FRDA ^f^ [12]	-	-	-	1	2	3	3

CMA: chromosomal microarray; MLPA: multiplex ligation-dependent probe amplification; NGS: next-generation sequencing (CES, WES, WGS); PCR: polymerase chain reaction; TP-PCR: triplet repeat primed PCR. The numbers indicate the sequential tier of genetic testing. Sanger sequencing is usually reserved for confirmation and analysis of family segregation. ^a^
*Dystrophinopathie*s. MLPA is the first tier genetic testing to detect exon deletion or duplication for Duchenne and Becker muscular dystrophies and other dystrophinopaties. NGS is the second tier test. ^b^
*Spinal muscular atrophy at chromosome 5q*. MLPA is the premier genetic test to detect exon 7 deletion of *SMN1* gene and the number of *SMN2* gene copies. In heterozygous patients, the second mutation is searched by either NGS or Sanger sequencing. ^c^
*Myotonic dystrophies 1 and 2*. Southern blotting of the *DMPK* and *CNBP* genes are used to determine the size of the CTG (DM1) and CCTG (DM2) dynamic repeat expansions. ^d^
*Facioscapulohumeral muscular dystrophies 1 and 2*. The length or number of repeat units of the *D4Z4* locus is classically determined by Southern blot analysis, typically with a probe (e.g., p13E-11) and localized immediately proximal to D4Z4 (double digestion with EcoRI and BlnI restriction enzymes to distinguish 4q35 and 10q26 regions). The complete approach to FSHD molecular diagnosis also requires the analysis of the methylation status of the *D4Z4* locus and sequencing of *SMCHD1* and *DNMT3B* genes. ^e^
*Charcot-Marie-Tooth 1A disease and inherited neuropathy with liability to pressure palsies*. Both CMT1A 1.4 Mb duplication and HNPP 1.4 Mb deletion at chromosome 17p11.2 are tested by either MLPA or CMA. ^f^ Friedreich ataxia. Southern blotting or long-range PCR of the *FXN* gene are used to determine the size of the GAA dynamic repeat expansion.

## Data Availability

Not applicable.

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
