# Peer review of "The Increasing Impact of Translational Research in the Molecular Diagnostics of Neuromuscular Diseases"

_ijms, 2021, doi:10.3390/ijms22084274_

Round 1

Reviewer 1 Report

The Authors report a comprehensive review on the impact that the translational research is acquiring in the molecular diagnosis of neuromuscular disorders. They describe all the approaches, ranging from the single-gene approach to the clinical exome sequencing (CES), WES and WGS, that are currently used stressing that, despite all efforts, about 40% of NMD patients remain undiagnosed.

They agree that, in order to reach a diagnostic result, a deep phenotypic characterization of the patient is essential, preferably with the use of the standardized Human Phenotypic Ontology (HPO). They also propose a Translational Diagnostic Program (TDP) they designed and adopted in house to facilitate the diagnostic process.

To this aim, a multiomics approach, implementing the translational diagnostic programs that combine the use of patient’s cell with that of animal models is strongly recommended.

The limitations and uncertainties of each approach are clearly described.

The Authors prospect a closer collaboration between all the actors of this scenario  – clinicians with expertise in NMD, geneticists, clinical scientists in diagnostic laboratories, pathologists and research based investigators - as well as the use of both approaches, translational research and omics, including experimental biology in the diagnostic process in centers of expertise and tertiary referral hospitals.

The paper is very interesting and of current interest. References are appropriate.

I have no further specific comments. I would just like to point out that on p. 10, after reference 26, the period included in lines 411-412 is repeated twice.

Author Response

We are grateful to the reviewer for the kind comments on the manuscript.

The repeated sentence on page 10, after reference 26 (now 38), included in lines 411-412 has been deleted.

Reviewer 2 Report

 The present  review is interesting and the experience provided by authors represent a good example of how genetic rare diseases may be studied and diagnosed. Indeed, the ideas and perspectives reported are not innovative but their applicability is interesting. Moreover there are issues to be addressed:

In the first part of the manuscript (Paragraphs 1-2) authors should cite more articles that employed the tecnologies they showed (e.g. MLPA, NGS...) for the diseases they reported (e.g. DMD/BMD, FSHD), such as Kong et al., 2019, PMID:31412794; Savarese et al., 2015, PMID: 25891276; Strafella et al., 2019 , PMID: 31600781..

Lines  411-415 are composed by a duplicated sentence.

The "Translational Diagnostics Program" is really exciting! However, it seems particularly useful to characterize VUS which represent exonic (above all missense variants), considering the reported functional studies. Authors should improve this section, by stating how they deal with variants within potential or established regulatory regions and how they study their effect on expression, which, as authors state, is notably tissue-specific. Moreover, how do they study (if so) mutations or variants that causes aberrant expression profiles specifically in muscular tissues? Indeed, there is a paragraph dedicated to RNAseq and transcriptional profiles, but it is not clear if authors include this kind of variants and these analyses within their translational diagnostic program.

Author Response

The present review is interesting and the experience provided by authors represent a good example of how genetic rare diseases may be studied and diagnosed. Indeed, the ideas and perspectives reported are not innovative but their applicability is interesting. Moreover, there are issues to be addressed:

  1. In the first part of the manuscript (Paragraphs 1-2) authors should cite more articles that employed the technologies they showed (e.g. MLPA, NGS...) for the diseases they reported (e.g. DMD/BMD, FSHD), such as Kong et al., 2019, PMID:31412794; Savarese et al., 2015, PMID: 25891276; Strafella et al., 2019 , PMID: 31600781.

We thank the reviewer for this comment and suggestions. We have incorporated more cites in both the Table and the main text: (a) in the Table we have added references 6 to 12 for the disease-specific tests that address the molecular diagnosis; (b) regarding NGS-based genetic testing, we now are mentioning several references (13 to 16) in the main text, page 2, lines 83-84.

Consequently, the entire reference numbering has been changed.

  1. Lines  411-415 are composed by a duplicated sentence.

We thank the reviewer’s comments on our manuscript. These lines have been removed.

  1. The "Translational Diagnostics Program" is really exciting! However, it seems particularly useful to characterize VUS which represent exonic (above all missense variants), considering the reported functional studies. Authors should improve this section, by stating how they deal with variants within potential or established regulatory regions and how they study their effect on expression, which, as authors state, is notably tissue-specific.

We are grateful to the reviewer for the comments on this point. We have included the following on page 9, lines 372-381: “We have presented examples where the VUS is a missense variant. It should be noted that TDP can also be applied in the case of genetic variants that affect consensus splice regions and even regulatory regions of gene expression. For VUS affecting canonical splice sites, a splicing study can be performed using minigenes by cloning the genomic region [66] or by analyzing RNA from a patient's fibroblasts by RT-PCR. These two strategies examine the impact of a given VUS on the splicing and also provide information on the transcripts affected by the variant. For VUS of regulatory regions, cloning of the genomic region for studies of luciferase activity could be an option for the evaluation of the functional impact on the transcriptional control of gene expression.”

  1. Moreover, how do they study (if so) mutations or variants that causes aberrant expression profiles specifically in muscular tissues? Indeed, there is a paragraph dedicated to RNAseq and transcriptional profiles, but it is not clear if authors include this kind of variants and these analyses within their translational diagnostic program.

We have included comments on RNA sequencing in the ‘Multi-omics approaches section’. In the paragraph between current lines 414 to 418, we state “RNA sequencing allows detecting aberrant expression, aberrant splicing and mono-allelic expression. Some of the challenges of using this technology reside in sample variability due to different factors. For instance, tissue selection for RNA sequencing is particularly relevant because gene expression is expected to be different according to the disease target tissue.”

RNA-seq can be a very useful tool to detect some pathogenic variants that are not detected in exome approaches. Since the TDP focuses on functional studies, RNA-seq is not currently considered to be part of it; rather, it should be considered in the clinical genomics services for some selected cases.

Round 2

Reviewer 2 Report

Authors correctly answer to the questions and properly revised the manuscript  as requested.